# Efficiency of an Air Cleaner Device in Reducing Aerosol Particulate Matter (PM) in Indoor Environments

**DOI:** 10.3390/ijerph17010018

**Published:** 2019-12-18

**Authors:** Paola Fermo, Valeria Comite, Luigi Falciola, Vittoria Guglielmi, Alessandro Miani

**Affiliations:** 1Dipartimento di Chimica, Università degli Studi di Milano, 20133 Milano, Italy; 2SIMA (Società Italiana di Medicina Ambientale), Via Monte Leone 2, 20149 Milano, Italy; 3Dipartimento di Scienze e Politiche Ambientali, Università degli Studi di Milano, 20133 Milano, Italy

**Keywords:** air cleaner, particulate matter, indoor air quality

## Abstract

Indoor air quality (IAQ) in household environments is mandatory since people spend most of their time in indoor environments. In order to guarantee a healthy environment, air purification devices are often employed. In the present work, a commercial household vacuum cleaner has been tested in order to verify its efficiency in reducing the mass concentration and particle number of aerosol particulate matter (PM). The efficiency has been tested measuring, while the instrument was working, PM_10_ (particles with aerodynamic diameter less than 10 μm), PM_2.5_ (particles with aerodynamic diameter less than 2.5 μm), PM_1_ (particles with aerodynamic diameter less than 1 μm), and 7 size-fractions in the range between 0.3 and >10 μm. Measurements have been carried out by means of a portable optical particle counter instrument and simulating the working conditions typical of a household environment. It has been found that the tested commercial device significantly reduces both PM concentrations and particle number, especially in the finest fraction, i.e., particles in the range 0.3–0.5 μm, allowing an improvement of indoor air quality.

## 1. Introduction

Air pollution is the largest single environmental risk to health, responsible for an estimated 7 million premature deaths every year globally and about 556,000 in the European Region.

As people spend a considerable amount of time indoor, either at work or at home, indoor air quality (IAQ) plays a significant part in their general state of health, wellbeing, and human performances [1]. This is particularly true for children, elderly people, and other vulnerable groups.

Household indoor air pollution is one of the goals of the 2030 Agenda for Sustainable Development (https://ec.europa.eu/environment/sustainable-development/SDGs/index_en.htm), and, in particular, the Sustainable Development Goals (SDGs) that are related to the topics of health, sustainable cities, industrialization, and mitigating the effects of climate change, are numerous.

IAQ in household environments is therefore mandatory in private homes, schools, workplaces, etc. Inside houses, offices, etc., there are many indoor sources ranging from cooking to domestic works [2,3]. In particular, house cleaning operations are one of the main sources since generate indoor pollutants in both gaseous and particle phases. Among the house cleaning operations, the use of vacuum cleaners could represent a potential source of health risk due to the enormous quantity of particles that are dispersed during its use. Due to the increase of allergic rhinitis, the utility of indoor environmental management deserves increasing attention [4] and the use of air filtration systems is spreading [5].

Aerosol particulate matter (PM) is among the main pollutants both outdoor and indoor. PM_10_ (particles with aerodynamic diameter less than 10 μm), PM_2.5_ (particles with aerodynamic diameter less than 2.5 μm), and PM1 (particles with aerodynamic diameter less than 1 μm) are the most harmful air pollutants together with ground-level ozone, nitrogen oxides, and sulfur oxides that are also of concern being the main precursors of secondary PM (particulate matter) in the atmosphere.

Despite reductions in emissions of PM_10_, the majority (50–92%) of the urban population in monitored European countries was exposed to concentrations above the air quality goal.

For example, levels of PM_2.5_ exceeded the goal established by WHO at 75% of stations in European Region in 2015, and exposure to PM reduces the life expectancy of every person by an average of almost 1 year, mostly because of the increased risk of cardiovascular and respiratory diseases, and lung cancer, with an enormous cost of the health care with a prevalence of diseases among children and adolescents. Exposure to air pollution has been associated with numerous health effects that range from minor upper respiratory irritation to chronic respiratory and heart disease, lung cancer, and acute respiratory infections [6]; in particular, exposure has been associated with respiratory infections, asthma and allergic symptoms, ear inflammation, and deficits in lung function, as well as cognitive impairment [6]. Exposure to PM during pregnancy has also been associated with adverse to birth outcomes, such as preterm births and low birthweights [7].

The negative effects of PM on human health are linked, as it is well known from the literature, not only to the particles’ concentration but also to their chemical composition, which should be considered [6,8] in order to highlight health relevant components, including the measurements of organic and elemental carbon [9]. A large numbers of studies have been carried out on this topic also focusing on sources identification [10,11,12], size-resolved measurements [13], and on specific markers investigation [8,14,15].

As far as indoor environments are concerned, there are no limits for air quality, even though we spend about 80% of our time indoors [1]. The U.S. Environmental Protection Agency (EPA) has shown that the quality of indoor air is 5–10 times worse than outdoor because of both penetration of pollutants from outdoor and, as stated before, because of the presence of specific indoor sources. In China, heavy pollution episodes are frequent and strongly influence indoor air quality [16].

Our homes are full of particles including also dust mites, which are one of the most common type of indoor allergens and the most important factor in causing asthma in children [17].

Due to poor IAQ, the use of air purification devices, like that tested in the present paper, is increasing worldwide [5], even though, ventilating and air conditioning systems are often employed in many countries to achieve an improvement of indoor air quality.

In this study, a commercial household vacuum cleaner has been tested in order to verify its particle abatement capacity. The evaluation has been carried out using a portable optical particle counter instrument and simulating the working conditions typical of a household environment. Particle mass concentrations of PM_10_, PM_2.5_, PM_1_, and particle number concentration in 7 size-fractions between 0.3 and >10 μm have been measured.

## 2. Materials and Methods

In the present paper, a commercial vacuum cleaner that can also be used as air purification system has been tested. Traditional household vacuum cleaners for air purification generally have a filter or a dust bag as a barrier for dust collection. However, after use, collecting all the materials can stick to the surface of the filter or accumulate in the dust bag. The main drawback of this system is that dust can block or even damage the filter affecting the suction and the air flow rate.

The device tested in the present study is a water-based cleaning system and it can be used both as vacuum cleaner and air purification system.It can effectively filter dust particles and micro-organisms. Since the separator produces a strong centrifugal force, dust, dirt, and thin allergens can be separated in water and no filters or dust bags are required. The maximum suction flow rate is 156 m^3^/h.

The measurements were carried out by placing the device in parallel with a P-DustMonit (Contec, Milan, Italy) instrument inside a test chamber measuring 335 × 470 × 290 (H) cm (about 45 m^3^). The test chamber was ventilated with outdoor air before starting of the test, and, before proceeding with the start of the air purification device, it was expected that conditions were stabilized, i.e., that the trend of the curves relative to the three fractions of interest (PM_10_, PM_2.5_, and PM_1_) reached a plateau.

The P-DustMonit unit is a portable complete system for continuous monitoring of particulate concentration in the air. It is based on the light scattering principle: a constant flow pump draws air in through a radial symmetric probe where each particle is hit with a laser. The energy reflected by each particle, proportional to its dimension, is measured by a high-velocity photodiode, which generates counting signals corresponding to the particle’s dimensions. Instrument sample flow is 1 L/min, and the device is able to work in the temperature range −10–40 °C.

The instrument allows: (1) measurement of the concentrations of the fine particulate expressed as PM_10_–PM_2.5_–PM_1_ in μg/m^3^ (in real time and at the same time); (2) measurement in μg/m^3^ (in real time and simultaneously) of the concentrations of the inhalable, thoracic, and breathable dust, as defined by current regulations; (3) measurement of the particle number in real time classifying them simultaneously up to 15 different size classes.

During the experiment, the relative humidity (RH) and temperature were also registered by P-DustMonit unit—the temperature increased from about 22.5–24.1 °C while relative humidity (RH) decreased when air ventilation was performed.

P-DustMonit unit can be assumed as a valid system to track air quality. Optical particle count systems, like that one employed in this work, are often used to quantify fine and ultra-fine particles in both indoor and outdoor environments [3,18].

With regards to the intercomparison with other instruments, it is reported in the literature [18] that optical particle counters can be successfully used to analyze diurnal average trends in particle number and mass concentrations; furthermore, the high temporal resolution showed that optical detectors could be very useful for air quality applications and to investigate specific pollution events. However, especially, if they are used for evaluating mass concentrations (PM_1_, PM_2.5_, or PM_10_), it is worth to notice that it is necessary to take into account RH effects, even if the experimental conditions registered during our experiments were such that they were not a problem.

## 3. Results and Discussion

In order to test the vacuum cleaner’s efficiency as an air purification system, measurements have been carried out putting in parallel the device and the P-DustMonit unit inside a chamber without air exchange but not perfectly tight in order to simulate what can happen in a room of a house with the windows closed. The aim was to quantify both PM and particle sizes in different size-fractions.

PM_10_ concentrations inside the measurement chamber, after stabilization of the environmental conditions, was approximately 20 μg/m^3^. The device was started at the maximum speed corresponding to a suction flow rate of 156 m^3^/h. In Figure 1, it is clearly observable how PM concentrations (PM_10_, PM _2.5_, and PM_1_) rapidly decrease when the instrument is switched on at 10:20 a.m. The concentration of the three size-fractions after about 40 min has been reduced considerably, and, in particular, PM_10_ was reduced from 14 to about 7 μg/m^3^, and PM_2.5_ and PM_1_ from 13 to about 6 μg/m^3^.

At 10:40 a.m., a reduction of 50% of PM concentration has been achieved. It is also possible to hypothesize that by operating at the minimum speed (52 m^3^/h) in 2 h, it would be possible to achieve the same result that was achieved in 20 min.

At 11:25 a.m., the instrument was turned off and the room was ventilated by opening the windows. From the graph, it can be seen how, in particular, PM10 concentration rises very quickly.

Subsequently, at 11:38 a.m., after obtaining a stabilization of the concentrations, the system was again switched at the minimum speed (52 m^3^/h), and, after 20 min, it was observed that the concentration decreased from about 16 μg/m^3^ to about 12 μg/m^3^ with a reduction of 25%. Therefore, even when operating with the minimum speed, a significant decrease in PM is appreciated in a short time.

Furthermore, it is worth noting that a further advantage of the air purification system tested here is its long-term performance, thanks to particle abatement in water that allows to avoid worsening of performances due to dust loading, which normally happens for air cleaners based on filtration [19,20].

The measurement performed by P-DustMonit unit was carried out outdoor in the same time interval as indoor (i.e. between 10:20 and 11:40) in order to compare with the corresponding indoor PM values, and the outdoor recorded values showed some variations in PM concentrations mainly due to the influence of wind (Figure 2). As normally expected, indoor values are lower if the typical indoor sources are not present (cleaning operation, cooking, etc.). As a consequence, before vacuum cleaner was working, initial PM concentrations were about 10 μg/m^3^ lower than the outdoor values. Nevertheless, it is worth to notice that in our indoor experiment, the finer fractions (PM _2.5_ and PM_1_) are quite similar in concentration to PM10, while, in outdoor, the prevalent fraction is represented by PM_10_. It is worth noting that a lower PM_10_/PM_2.5_ (or PM_10_/PM_1_) ratio represents a greater health risk, with the finer fractions being more penetrating in the pulmonary alveoli.

Figure 3, Figure 4 and Figure 5 report the trends of the numerical concentrations of the particles within 7 size classes. The higher particles’ indoor concentration (more than 60,000 particles/L, i.e., 60 particles/cm^3^) have been registered for the fraction formed by particles with diameter >0.3 μm (Figure 5). In outdoor, the concentration level for the same fraction is double (with a maximum of 17,000 particles/L). The values registered in outdoor ambient air are in good accordance with data from the environmental protection Agency (ARPA) of Lombardy Region (Northern Italy) for Milan area [21], where our measurements have been carried out. Our values are also in fair accordance with the results obtained in other studies for the same particles’ dimensional range [3,22]. It is also interesting to notice that if ultra-fine particles (with diameter <0.1 μm) would be considered (this dimensional range cannot be measured with the instrument used in the present study, since P-DustMonit’s lower particle limit is 0.3 μm, an enormous increase in the number would have been observed [23] reaching values of the order of particle thousands per cm^3^.

Furthermore, it has been observed that UFP indoor concentrations are generally higher than outdoor ones because of the presence of numerous and intermittent indoor sources (cooking, cleaning operations, etc.) [3] with the consequence that people who spend most of the time at home have a high risk to be exposed to indoor pollutants in the finest fraction, i.e., the most dangerous ones.

From the comparison among the particle number trends reported in Figure 3, Figure 4 and Figure 5, it can be observed how the vacuum cleaner seem to be more efficient in reducing fraction > 0.3 µm. In order to make a more realistic comparison among the different size-fractions, within each size-fraction particle number was divided for the corresponding maximum value in that specific range. The results obtained are shown in Figure 6a,b depending on the dimensions. In addition, in this case, it is evident how the vacuum cleaner is more effective in the reduction of particles with diameter > 0.3 µm while when ventilation with outdoor air is performed, particles with diameter > 3 µm show the strongest change while a less significant decrease is observable for the other fractions. Therefore, the vacuum cleaner tested in this study has proved to be particularly efficient in reducing the number of finest particles, which is the greatest concern from the health point of view, being the one able to deeply penetrate the respiratory system until reaching the pulmonary alveoli.

So from a health point of view, since the finest particles are those that cause the most concern, their reduction in number is a very significant point to be achieved.

## 4. Conclusions

It has been demonstrated that the tested air cleaner can significantly reduce the indoor PM concentrations (PM_10_, PM_2.5_, and PM_1_) to values below WHO guideline level that correspond to 10 μg/m^3^. It is also worth noting that the particles that are more efficiently removed are those having the finest size, i.e., particles in the range 0.3–0.5 μm. PM being one of the priority causes that lead to the onset of diseases affecting the respiratory system, the use of such devices could contribute to significantly improving the household air quality. In fact, people that spend most of their time at homes or, in general, in indoor environments, may have a higher risk of acute or chronic exposure to typical indoor pollutants, and the possibility to use devices that allows a significant improvement of IAQ certainly contributes to improving health and wellbeing.

## Figures and Tables

**Figure 1 ijerph-17-00018-f001:**
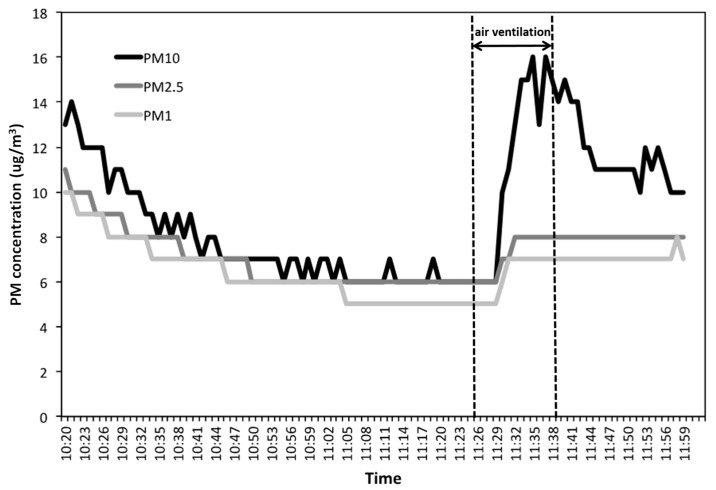
Trend of particulate matter 10 (PM_10_), PM_2.5_, and PM_1_ (μg/m^3^) fractions during the vacuum cleaner working (during air ventilation period, the device was stopped and then started again.)

**Figure 2 ijerph-17-00018-f002:**
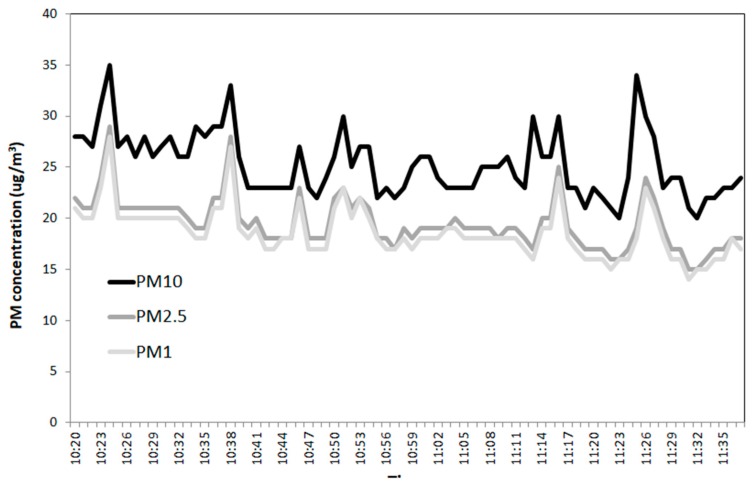
Trend of concentrations of PM_10_, PM_2.5_, and PM_1_ (μg/m^3^) measured outdoor.

**Figure 3 ijerph-17-00018-f003:**
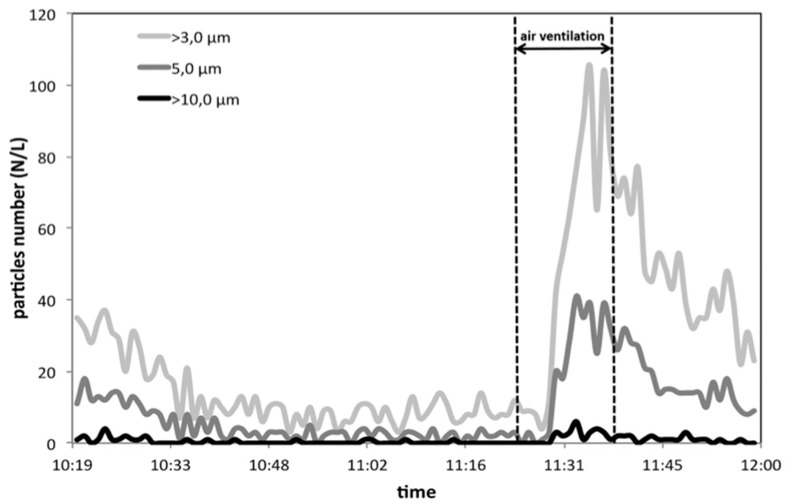
Trend of particle number per L for the fractions >3 μm, >5 μm, and >10 μm during the commercial vacuum cleaner working (during air ventilation period, the device was stopped and then started again.)

**Figure 4 ijerph-17-00018-f004:**
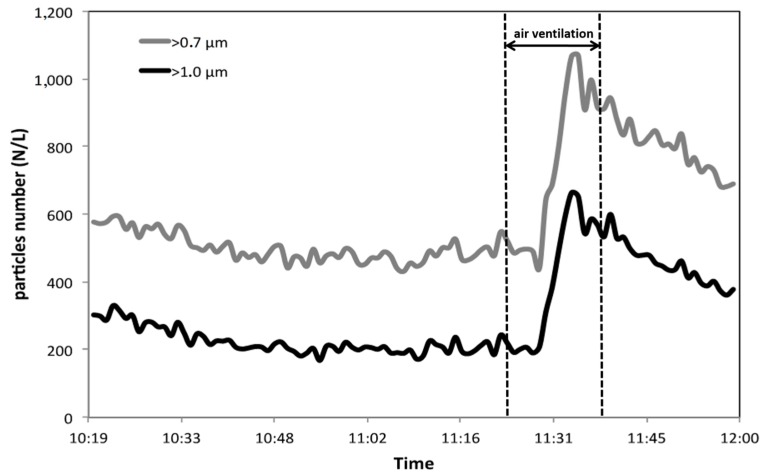
Trend of particle number per L for the fractions >1 µm and >0.7 µm during the use of commercial vacuum cleaner (during air ventilation period, the device was stopped and then started again.)

**Figure 5 ijerph-17-00018-f005:**
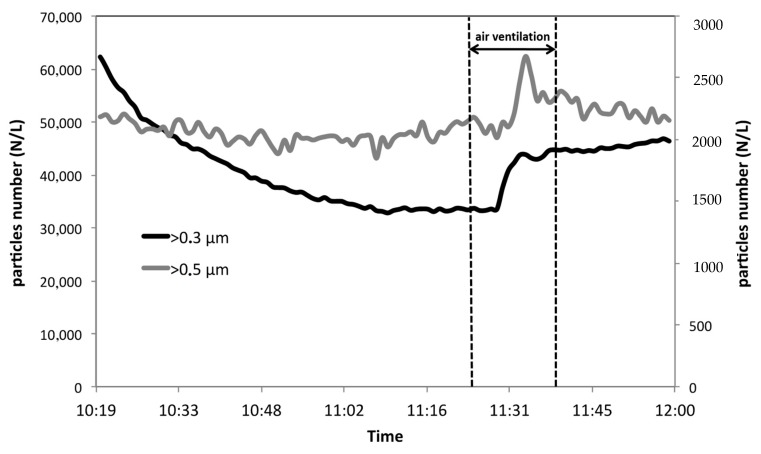
Trend of particle number per L for the fractions > 0.5 µm (secondary axis on the right) and > 0.3 µm (main axis on the left) during the use of commercial vacuum cleaner (during air ventilation period, the device was stopped and then started again.)

**Figure 6 ijerph-17-00018-f006:**
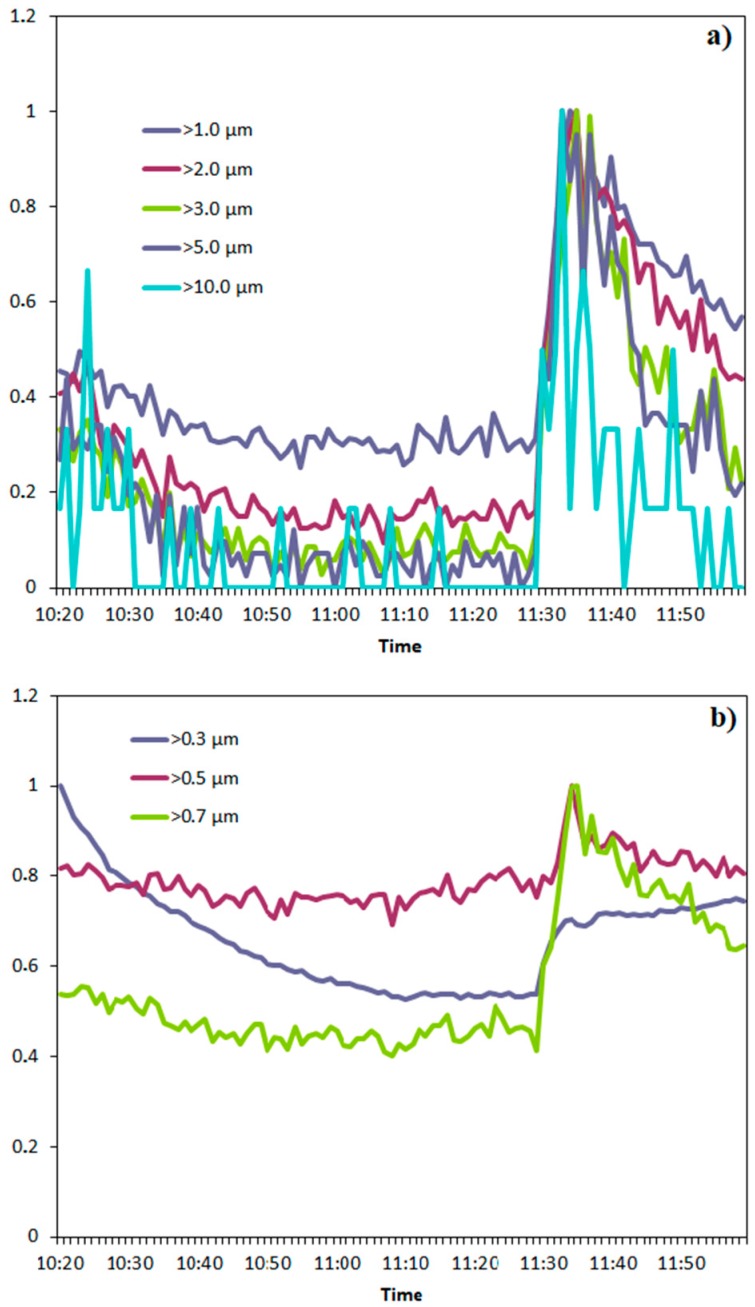
Trend of normalized particle number concentration for each fraction: (**a**) particles with diameter >1 µm; (**b**) particles in the range 0.3 µm < d < 0.7 µm.

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
