# Peer review of "Efficiency of an Air Cleaner Device in Reducing Aerosol Particulate Matter (PM) in Indoor Environments"

_ijerph, 2019, doi:10.3390/ijerph17010018_

Round 1

Reviewer 1 Report

Lines 64-65: It would be good to cite other sources of information, particularly outside of the US.  For example, relative outdoor PM concentrations to indoors in some cities in China potentially could be a variant, in episodes.

Lines 114-116: The hypothesis is not clearly explained.  It seems that the minimum speed was tested later, but It is not clear why this statement is made rather than testing it.  Would it be normal to run a vacuum for 2 hours?

Figures 1-4: The figures should include notation of the time periods during which the cleaner was running for improved clarity.

While the results are interesting and meaningful with regard to the need for reduction of airborne particulates in the indoor environment, the fact that only 1 cleaner was tested, under unspecified conditions (temperature, humidity), and that there was no comparison between this system and other similar types or conventional types of systems.  Thus, the usefulness of the results are limited.  A revision which includes such types of expanded results is encouraged.

Reviewer 2 Report

Dear Authors,

The manuscript presents a study of the evaluation of air purification system by using the specially designed vacuum cleaner. The general idea of the case study is interesting, since it provides an insight about air quality improvement by using the air purification system in the household. However the results are limited to relatively short case study, where due to the lack of measurements of the background levels of PM concentration, it is difficult to draw conclusions.

Therefore, I suggest to improve the study by repeating the measurements in different PM background conditions, which will allow to improve Discussion and conclusions.

Additionally, the Instrumentation section should be extended by providing more specific information on the instrumentation used for measurements.

Please find sepcific comments attached.

Round 2

Reviewer 2 Report

Dear authors,

The manuscript has been improved by implementing suggested corrections of the text and figures. However I still consider, that more extensive measurements (2 - 3 days of measurements instead of few hours) could better support conclusions.

I have some additional minor comments and corrections, especially regarding the number format in the Figures (change decimal comma to dot). I would also recommend English language and style editing before publication.

More comments are provided in the pdf.
